# Tracing the substrate translocation mechanism in P-glycoprotein

**Theresa Gewering[1,2†], Deepali Waghray[3†], Kristian Parey[1,2,4], Hendrik Jung[5], Nghi NB Tran[6], Joel Zapata[6], Pengyi Zhao[7], Hao Chen[7], Dovile Januliene[1,2,4], Gerhard Hummer[5,8], Ina Urbatsch[6], Arne Moeller[1,2,4*‡], Qinghai Zhang[3*‡]**

[1]Osnabrück University, Department of Biology/Chemistry, Structural Biology Section, Osnabrück, Germany; [2]Department of Structural Biology, Max Planck Institute of Biophysics, Frankfurt, Germany; [3]Department of Integrative Structural and Computational Biology, The Scripps Research Institute, La Jolla, United States; [4]Osnabrück University, Center of Cellular Nanoanalytic Osnabrück (CellNanOs), Osnabrück, Germany; [5]Department of Theoretical Biophysics, Max Planck Institute of Biophysics, Frankfurt, Germany; [6]Department of Cell Biology and Biochemistry, Texas Tech University Health Sciences Center, Lubbock, United States; [7]Department of Chemistry & Environmental Science, New Jersey Institute of Technology, Newark, United States; [8]Institute for Biophysics, Goethe University Frankfurt, Frankfurt, Germany

**\*For correspondence:**
arne.moeller@uos.de (AM);
qinghai@scripps.edu (QZ)

†These authors contributed equally to this work
‡These authors also contributed equally to this work

**Competing interest:** The authors declare that no competing interests exist.

**Abstract** P-glycoprotein (Pgp) is a prototypical ATP-binding cassette (ABC) transporter of great biological and clinical significance.Pgp confers cancer multidrug resistance and mediates the bioavailability and pharmacokinetics of many drugs (Juliano and Ling, 1976; Ueda et al., 1986; Sharom, 2011). Decades of structural and biochemical studies have provided insights into how Pgp binds diverse compounds (Loo and Clarke, 2000; Loo et al., 2009; Aller et al., 2009; Alam et al., 2019; Nosol et al., 2020; Chufan et al., 2015), but how they are translocated through the membrane has remained elusive. Here, we covalently attached a cyclic substrate to discrete sites of Pgp and determined multiple complex structures in inward- and outward-facing states by cryoEM. In conjunction with molecular dynamics simulations, our structures trace the substrate passage across the membrane and identify conformational changes in transmembrane helix 1 (TM1) as regulators of substrate transport. In mid-transport conformations, TM1 breaks at glycine 72. Mutation of this residue significantly impairs drug transport of Pgp in vivo, corroborating the importance of its regulatory role. Importantly, our data suggest that the cyclic substrate can exit Pgp without the requirement of a wide-open outward-facing conformation, diverting from the common efflux model for Pgp and other ABC exporters. The substrate transport mechanism of Pgp revealed here pinpoints critical targets for future drug discovery studies of this medically relevant system.

## eLife assessment

P-glycoprotein is a major ABC-transporter that exports drugs used in chemotherpay and effects the pharmacokinetics of other drugs. Here the authors have determined cryo-EM structures of drug complexes in previously unforeseen outward-facing conformations. These **convincing** findings are mechanistically **important** and reveal potential regions to be exploited by rational-based drug design.

## Introduction

P-glycoprotein (Pgp, also known as MDR-1 and ABCB1) is a prominent member of the human ATP-binding cassette (ABC) efflux transporters that removes a large variety of chemically unrelated hydrophobic compounds from cells (*Sharom, 2011*; *Chufan et al., 2015*). Pgp is highly expressed in the liver, kidney, and intestines, where it limits absorption and enhances the excretion of cytotoxic compounds (*Thiebaut et al., 1987*; *Cascorbi, 2011*; *Cordon-Cardo et al., 1990*). In the blood-brain, blood-testis, and blood-placenta barriers, Pgp protects and detoxifies sanctuaries from xenobiotics (*Schinkel, 1999*; *Borst and Schinkel, 2013*; *Fromm, 2004*). Of central importance for many chemotherapeutic treatments, Pgp is a major determinant of drug bioavailability and pharmacokinetics and confers multidrug resistance in several diseases, most notably cancer (*Szakács et al., 2006*; *Gottesman and Ling, 2006*; *Giacomini et al., 2010*). Consequently, evasion, selective inhibition, and modulation of Pgp transport are important goals in drug development, but are hindered by a lack of detailed understanding of the drug transport mechanisms (*Waghray and Zhang, 2018*).

For Pgp and ABC transporters in general, ATP binding, hydrolysis, and subsequent release of the cleaved products (inorganic phosphate and ADP) fuel large-scale conformational changes that ultimately result in translocation of substrates across the membrane bilayer (*Rees et al., 2009*; *Hofmann et al., 2019*). In inward-facing (IF) conformations, the nucleotide-binding domains (NBDs) are separated, and substrates can access a large binding cavity that is open to the lower membrane leaflet and cytoplasm. ATP-induced dimerization of the NBDs promotes large rearrangements of the transmembrane domains (TMDs) from an IF to an outward-facing (OF) conformation, from which the substrate is released (*Januliene and Moeller, 2020*; *Verhalen et al., 2017*). Pgp structures with bound substrates and inhibitors in IF conformations have revealed one or more overlapping binding sites (*Aller et al., 2009*; *Loo and Clarke, 2000*; *Loo et al., 2009*; *Alam et al., 2019*; *Nosol et al., 2020*; *Szewczyk et al., 2015*; *Alam et al., 2018*). Multimodal binding mechanisms for chemically related compounds have been proposed (*Aller et al., 2009*; *Chufan et al., 2015*), including access of hydrophobic ligands from the lipid-protein interface to the binding sites of Pgp (*Nosol et al., 2020*; *Szewczyk et al., 2015*). However, structural data that elucidate how substrates are shuttled across the lipid bilayer to the extracellular space are unavailable at present. So far, only a single OF structure of mammalian Pgp has been described, but without detectable substrate density (*Kim and Chen, 2018*). Most insights into the transport pathway arise from molecular dynamics (MD) simulations, which are also limited by the scarcity of available structural details (*Göddeke and Schäfer, 2020*; *McCormick et al., 2015*). Consequently, the molecular mechanics that move the substrate through the transmembrane passage have remained elusive. To address this fundamental question and overcome the dynamic and transient nature of substrate translocation, we tethered a substrate molecule covalently to specific residues along the putative translocation pathway of Pgp and utilized cryoEM to capture Pgp structures with bound substrate during various stages of transport.

## Results

### Covalent ligand design for Pgp labeling

For covalent labeling of Pgp, we synthesized a derivative of the previously published QZ-Ala tripeptide substrate (*Szewczyk et al., 2015*), by substituting one of the three Ala with Cys that is disulfide-linked to a 2,4-dinitrophenyl thiolate group (designated AAC-DNPT, see Materials and methods, *Figure 1—figure supplement 1*). QZ-Ala is a strong ATPase stimulator and subject to Pgp transport as evidenced by its reduced cytotoxicity in cells overexpressing Pgp (*Szewczyk et al., 2015*). Additionally, we have measured Pgp-mediated transport of QZ-Ala in the MDCK-ABCB1 monolayer permeability assay (see Materials and methods). The efflux ratio ($R_E$) of QZ-Ala was determined as 2.5. In the presence of the Pgp inhibitor cyclosporin A, $R_E$ was 0.9, indicating that the Pgp-specific transport of QZ-Ala was inhibited. Derivatizing QZ-Ala is simplified by its structural symmetry, which reduces the complexity of modifying this transport substrate for attachment to Pgp in different states. Furthermore, attachment of DNPT strongly activates disulfide exchange, allowing rapid and efficient crosslinking between the cyclic peptide and accessible cysteines. Free DNPT, released during crosslinking, is bright yellow. This provides a simple, visible readout that can be monitored with a UV-Vis spectrometer (*Figure 1a*). Four single-Cys mutants of Pgp (Pgp335, Pgp978, Pgp971, and Pgp302) were generated for crosslinking, and mutations were chosen near the two previously reported QZ-Ala binding sites (*Szewczyk*

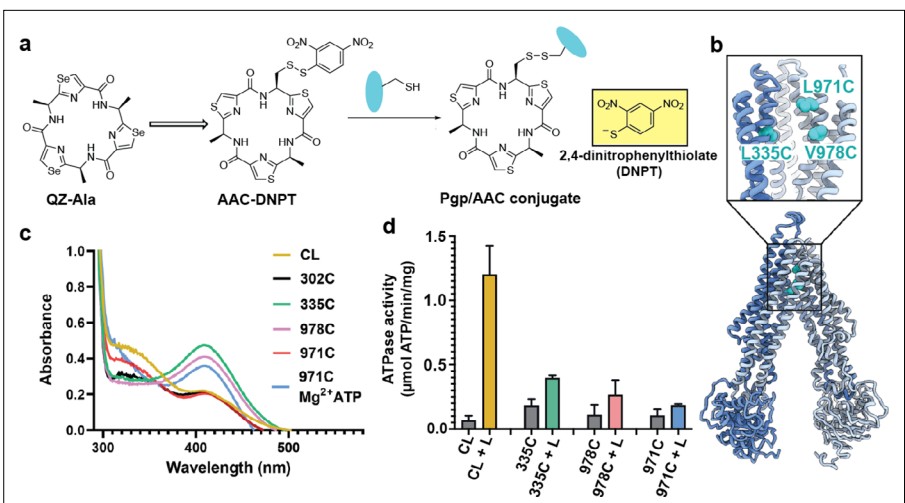

**Figure 1.** Covalent ligand design for P-glycoprotein (Pgp) transport studies. (**a**) Design of AAC-DNPT, a derivative of the cyclic peptide QZ-Ala, for disulfide crosslinking with single-cysteine mutants of Pgp; yellow product DNPT used as readout. (**b**) Positions of L335 (TM6), V978, and L971 (TM12) in Pgp, residues targeted for cyclic peptide labeling. (**c**) Monitoring covalent reaction between AAC-DNPT and cysteine mutants (±Mg$^{2+}$ ATP) by UV-visible spectrometry; CL-Pgp and I302C were included as negative controls. (**d**) ATPase activities of L335C, V978C, and L971C before and after crosslinking with AAC-DNPT; CL-Pgp was used as a control. Data are averaged from triplicate measurements with standard error bars.

The online version of this article includes the following figure supplement(s) for figure 1:

**Figure supplement 1.** Chemical synthesis of AAC-DNPT.

**Figure supplement 2.** MS detection of 335C peptide before and after AAC labeling.

**Figure supplement 3.** MS detection of 978C peptide before and after AAC labeling.

**Figure supplement 4.** MS detection of 971C peptide before and after AAC labeling.

**Figure supplement 5.** ATPase activities of P-glycoprotein (Pgp) mutants L335C, V978C, and L971C.

---

*et al., 2015*). L335C in transmembrane helix 6 (TM6) and V978C in TM12 are situated at symmetric positions on opposing TMD halves, while L971C is located two helical turns above V978 (*Figure 1b*). Residues L335C, V978C, as well as I302C located in TM5 were previously shown to be labeled with verapamil (*Loo and Clarke, 2001*). Based on the appearance of the yellow DNPT byproduct, Pgp335 and Pgp978 reacted with AAC-DNPT within minutes, while Pgp971 and Pgp302 did not. Even after extended incubations only background absorbances were detected comparable to Cys-less Pgp (CL-Pgp) (*Figure 1c*). Only after adding Mg$^{2+}$ ATP to fuel substrate translocation, Pgp971 reacted with AAC-DNPT, but Pgp302 did not (data not shown). Covalent attachment of the cyclic peptide AAC was further validated by high-resolution mass spectrometry of trypsin-digested peptide fragments (*Figure 1—figure supplements 2–4*). The presence of non-covalent transport substrates such as QZ-Ala or verapamil increase ATP hydrolysis in CL-Pgp and the single Cys mutants (*Figure 1—figure supplement 5a*). Similar to QZ-Ala, AAC-DNTP stimulated ATPase activity of CL-Pgp at low concentrations; with half-maximal enhancement of activity (EC50) seen at submicromolar concentrations, suggesting that it also acts as substrate (*Figure 1—figure supplement 5b*). At higher concentrations, the rate of ATP hydrolysis is significantly reduced in the mutants compared to CL-Pgp indicative of crosslinking AAC-DNPT with the single Cys mutants (*Figure 1d* and *Figure 1—figure supplement 5b*). Importantly, Pgp335 and Pgp978 retain notably higher ATPase activity after crosslinking than the respective apo-Pgp mutants in the absence of substrate (*Figure 1—figure supplement 5b* and *Figure 1d*), suggesting that tethered AAC still acts as a substrate and that Pgp335 and Pgp978 cycle between IF and OF conformations.

## Pgp structures with bound substrates

To capture different conformations during the transport cycle of Pgp, we mutated the catalytic glutamate residues in the Walker-B motifs of each NBD (E552Q/E1197Q) in Pgp335 and Pgp978. When

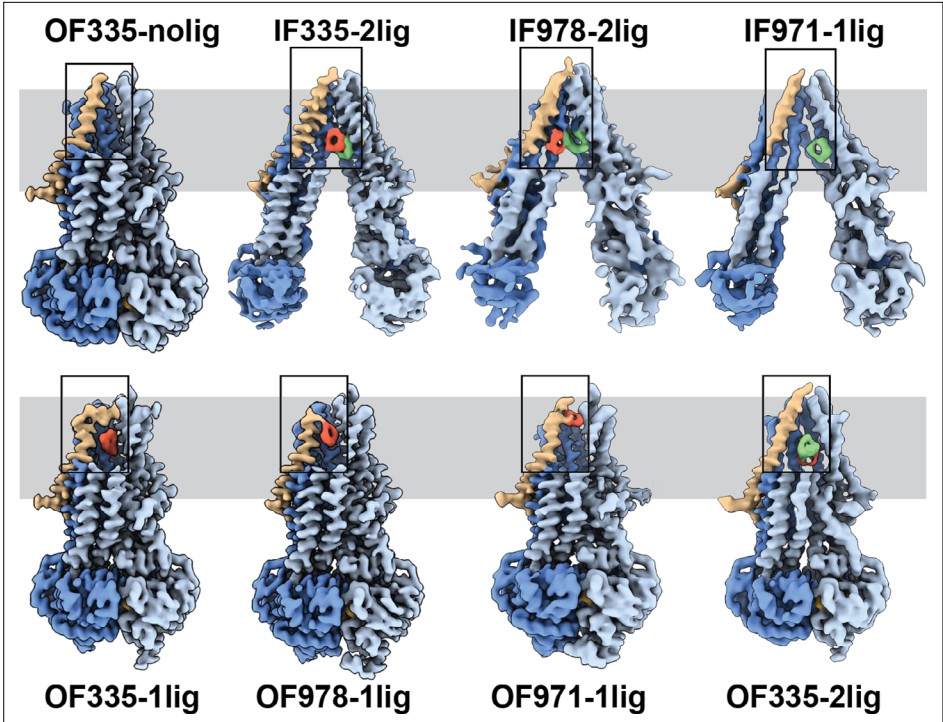

**Figure 2.** High-resolution structures of P-glycoprotein (Pgp) with bound ligands. For all mutants, inward-facing (IF) and outward-facing (OF) structures were obtained, with ligands revealed in the transmembrane region (gray shadow). For Pgp335 three different OF conformations were detected in one dataset (OF335-nolig, OF335-1lig, OF335-2lig). To display the binding sites, the upper halves of TM2,10,11, and 12 were removed from the densities. For the IF structures, TM9 was also removed for clarity. TM1 is shown in brown, with the covalently bound ligand in red and the non-crosslinked in green. Significant conformational changes were observed among these structures from which a further zoom-in on TM1 (black box region) was shown in *Figure 4* and *Figure 2—figure supplement 3*.

The online version of this article includes the following figure supplement(s) for figure 2:

**Figure supplement 1.** Summary of cryoEM data analysis for all structures.

**Figure supplement 2.** High-resolution features of the outward-facing (OF) structures.

**Figure supplement 3.** Comparison of the binding cavities of all structures from the crosslink experiments.

**Figure supplement 4.** Comparison of AAC binding in the different P-glycoprotein (Pgp) structures.

**Figure supplement 5.** Example of cryoEM data processing pathway for P-glycoprotein (Pgp).

**Figure supplement 6.** Visualization of tunnels in (**a**) outward-facing and (**b**) inward-facing P-glycoprotein (Pgp) structures by the CAVER plugin for PyMOL (*Chovancova et al., 2012*).

**Figure supplement 7.** Residues in P-glycoprotein (Pgp) mutants interacting with AAC and AAC-DNPT.

**Figure supplement 8.** Comparison of OF335 with one AAC bound (blue) compared to two AAC molecules bound (gray).

---

Mg$^{2+}$ ATP is present, these mutations arrest the enzyme in an ATP-occluded, NBD-dimerized conformation that resembles the pre-hydrolysis state (*Kim and Chen, 2018*). For Pgp971, we prepared the covalent complex in the presence of Mg$^{2+}$ ATP and stabilized the resulting OF complex with vanadate in a post-hydrolysis intermediate state (*Urbatsch et al., 1995a*; *Moeller et al., 2015*; *Figure 1c*). CryoEM analysis of the respective covalent complexes in both OF and IF conformations provided eight high-resolution structures with zero, one, or two ligands bound. In addition, a control dataset with ATP, but without the substrate, was collected for Pgp335 (*Figure 2*, *Figure 2—figure supplements 1 and 2*, and *Supplementary files 1 and 2*).

The IF structure of Pgp335, solved at 3.8 Å resolution, shows clear densities for two substrate molecules: one crosslinked and one not (IF335-2lig, *Figure 2* and *Figure 2—figure supplements 2 and 3*). The non-crosslinked molecule is bound in a position that resembles the one previously reported

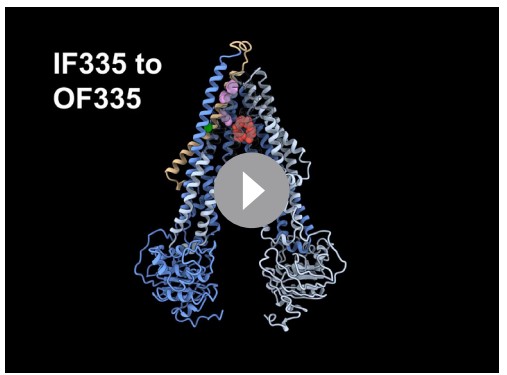

**Video 1.** Morph between the individual models illustrates the sequence of conformational changes during substrate translocation from IF335 to OF335. https://elifesciences.org/articles/90174/figures#video1

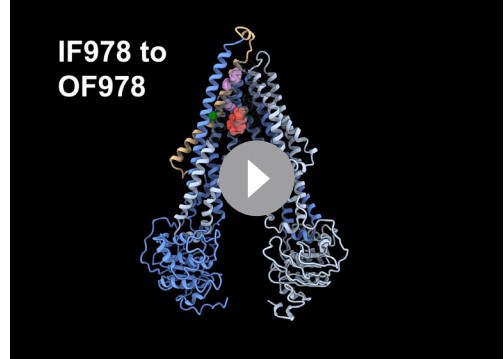

**Video 2.** Morph between the individual models illustrates the sequence of conformational changes during substrate translocation from IF978 to OF978. https://elifesciences.org/articles/90174/figures#video2

in a crystal structure (*Szewczyk et al., 2015*). It interacts closely with the covalently linked substrate, which is situated further up the membrane. Both ligand molecules are oriented along the vertical axis of the protein and positioned side-by-side through hydrophobic interactions. The uncleaved DNPT group of the non-crosslinked substrate is evident in the cryoEM density map. Two substrate molecules are also bound in the IF structure of Pgp978 (IF978-2lig); the non-covalently bound molecule is clearly visible at a similar position as in IF335-2lig, while the crosslinked substrate is located on the opposite side of the binding cavity near TM12 (*Figure 2—figure supplement 3*). Corroborating our labeling data (*Figure 1c*), in the IF structure of Pgp971 (IF971-1lig) at 4.3 Å resolution, only the non-crosslinked substrate is bound at the approximate location seen for non-crosslinked substrate in IF335-2lig and IF978-2lig (*Figure 2*, *Figure 2—figure supplements 3 and 4e*).

Multi-model cryoEM analyses of ATP-bound Pgp335 revealed three different OF conformations within a single dataset: ligand-free (40.4%), single-ligand-bound (33.3%), and double-ligand-bound (26.4%) (OF335-nolig, OF335-1lig, and OF335-2lig), at 2.6 Å, 2.6 Å, and 3.1 Å resolution, respectively (*Figure 2—figure supplement 5*, *Figure 2*). For Pgp978 and Pgp971, only OF conformations with a single bound ligand (OF978-1lig and OF971-1lig, *Figure 2*) were captured at 2.9 Å and 3 Å resolution, respectively.

## Substrate translocation pathway

In the OF structures with a single bound substrate, the central transmembrane helices TM1, TM6, TM7, and TM12 bulge outward to accommodate the substrate in a small tunnel (*Figure 2—figure supplement 4*). Comparing the ligand positions between the IF and OF structures, in IF335-2lig to OF335-1lig, and IF978-2lig to OF978-1lig, the covalently attached ligand pivots around the respective Cys residue by almost 180°, moving it further up the translocation tunnel (*Figure 2—figure supplements 3 and 4a, b*, and *Videos 1 and 2*). Our data do not reveal whether rotation of the ligand during the IF-OF conformational change is required for transport or if this reflects the fact that the ligand is covalently pinned to the crosslinking residue. However, the observation of ligand rotation illustrates how much space is available in the binding cavity during this conformational transition. The ligand position and orientation in OF335-1lig and OF978-1lig overall coincide well, with the latter shifting up slightly (*Figure 2—figure supplement 4d*). This finding indicates that the export tunnel is inherent to substrate translocation regardless of the initial binding position in the IF conformation. As such, our data suggest a single common pathway for substrate export in Pgp, contrary to the dual pseudosymmetric pathways suggested earlier (*McCormick et al., 2015*; *Parveen et al., 2011*).

OF971-1lig captures a later-stage transport intermediate, in which the compound resides approximately 7 Å further up the translocation tunnel and is coordinated by highly conserved residues, including M74 and F78 (*Figures 2 and 3b*, *Figure 2—figure supplements 3 and 7*). As no crosslinked substrate was detected in the corresponding IF structure, the substrate was likely transported to residue L971C before the crosslinking reaction could take place. Substrate entry from the extracellular

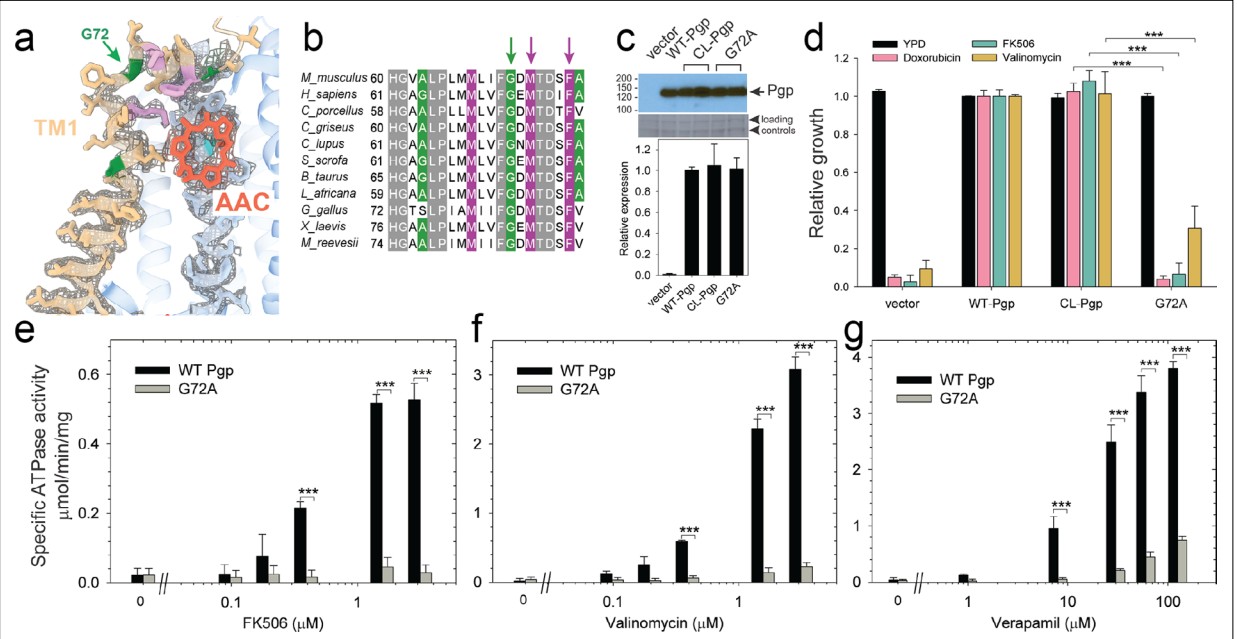

**Figure 3.** Functional characterization of G72A mutant. (**a**) TM1 breaks at the G72 position in the mid-transport states of P-glycoprotein (Pgp). Shown is a close-up view of TM1 (brown) with side chain densities, the G72 position (green arrow), and the adjacently bound cyclic peptide AAC (red sticks) in the OF335-1lig structure. (**b**) Sequence alignment of TM1 in vertebrate Pgps including the highly conserved G72 residue (green arrow). (**c**) Level of expression of the G72A mutant relative to the WT- and CL-Pgp in *Saccharomyces cerevisiae* cultures derived from the western blot analysis. (**d**) In vivo activity of the G72A mutant. The growth resistance of *S. cerevisiae* cultures transformed with either vector control plasmid, WT-Pgp, CL-Pgp, or the G72A mutant against several fungicides (doxorubicin, FK506, and valinomycin) relative to the YPD medium only (black bar) were analyzed. Compared to WT-Pgp and CL-Pgp, the G72A mutant severely compromised growth resistance against the tested drugs (p-values of <0.001 are indicated by ***). (**e–g**) ATPase activity of the purified G72A mutant. While basal ATP hydrolysis rates (in the absence or at low concentrations of drug) were indistinguishable between WT and G72A mutant proteins, stimulation of the ATPase activities by 1–3 µM FK506, 1–3 µM valinomycin, or 30–130 µM verapamil were severely impaired (p-values of <0.001 are indicated by ***).

The online version of this article includes the following figure supplement(s) for figure 3:

**Figure supplement 1.** Processing overview for OF335-G72.

side is less likely given the small opening of the substrate translocation pathway in all OF Pgp structures with or without bound substrate obtained here and previously (*Kim and Chen, 2018*). In all substrate-bound OF conformations, methionines and phenylalanines at different positions seem to be important for ligand coordination in all of the Pgp structures (*Figure 2—figure supplement 7*). As expected, the substrate is held in the binding pocket mainly by the hydrophobic interactions throughout the different conformations (*Figure 2—figure supplements 3 and 7*). In the OF335-nolig structure, as well as in the virtually indistinguishable control, the translocation tunnel is collapsed, which shields the substrate binding pocket from the extracellular milieu and prevents re-entry from this side (*Kim and Chen, 2018*).

## Regulatory role of TM1

The series of IF and OF Pgp structures reveal a cascade of conformational changes in TM1, which displays significant plasticity throughout the different transport stages (*Figure 2*, *Figure 2—figure supplement 3*, and *Videos 1–3*). In IF conformations, TM1 is a long, straight helix (*Figure 2* and *Figure 2—figure supplement 3*). However, in OF335-1lig, OF978-1lig, and OF971-1lig, TM1 swings out at A63, dilating the transmembrane passage and providing sufficient space for accommodating the bound ligand. At the same time, a pronounced kink at G72 on TM1 destabilizes the helix and the extracellular loop of Pgp between TM1 and TM2 and leads to a partial unwinding. This conformational deformation shields the intramembranous tunnel from the extracellular environment, with TM1 acting as a lid, almost parallel to the membrane (*Figures 2 and 4a*, *Figure 2—figure supplement 3*). Strikingly, along with the upward movement of the ligand (from L335C to V978C to L971C), the TM1 loop

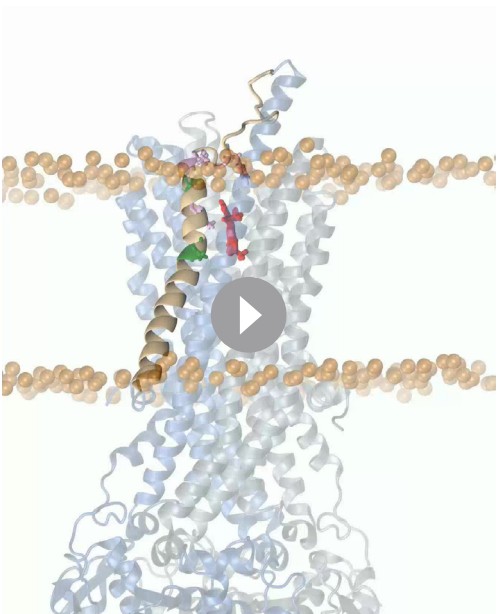

**Video 3.** Molecular dynamics (MD) simulation of substrate movement and release from OF978. https://elifesciences.org/articles/90174/figures#video3

is progressively lifted and restabilized. Ultimately, as shown previously (*Kim and Chen, 2018*) and by our OF335-nolig structure, the helix is straightened upon the release of the ligand (*Figure 4a*, *Figure 2—figure supplement 3*). Interestingly, the extracellular gate is sealed in all OF structures, which is especially surprising for OF971-1lig where the ligand is near the tunnel exit (*Figure 2*, *Figure 2—figure supplement 6*).

Using OF978-1lig or OF335-1lig as initial configuration, we performed MD simulations of Pgp embedded in a lipid bilayer to validate the proposed substrate escape pathway. In simulations starting from OF978-1lig at T=400 K, we observed movement and release of the non-covalently bound substrate to the extracellular side moderated by interactions between the ligand and hydrophobic residues in the upper part of TM1 (*Video 3*). The intermediate steps coincided well with the conformations observed in cryoEM (*Figure 4*, *Figure 4—figure supplements 1 and 2*). Without any bias imposed, the substrate progressed to a position closely matching OF971-1lig (*Figures 2 and 4b*). Then, while TM1 gradually refolded from the intracellular side, the substrate wiggled out to form transient interactions with the still unstructured extracellular part of TM1 (*Figure 4*). Importantly, the substrate escaped without Pgp experiencing a wide opening of the extracellular region. After the substrate left the exit tunnel, TM1 continued to refold to attain a straight helix configuration like OF335-nolig at the end of the simulation (*Figure 4*). In a simulation starting from OF335-1lig performed at T=330 K, a similar release pathway was observed (*Figure 4—figure supplement 3*). As the only difference, the substrate has not fully dissociated from the protein after 2.5 μs, with weak interactions persisting at the top part of TM1 from the extracellular side. Importantly, this is a configuration observed also in higher temperature simulations but with much shorter lifetime. Overall, with exception of TM1, motions in Pgp to accommodate the moving substrate were mostly limited to small dilations of the transmembrane helices, supporting our observations from cryoEM.

Our cryoEM structures and MD simulations revealed an important regulatory role of TM1 during substrate transport, emphasizing the helix-breaking G72. Therefore, we mutated this residue, which is highly conserved in mammalian Pgp (*Figure 3*), to a helix-stabilizing alanine (*López-Llano et al., 2006*) and tested the mutant's ability to export structurally diverse fungicides and to convey drug resistance in vivo. Compared to the wild-type and CL-Pgp, the G72A mutant expressed normally in *S. cerevisiae* (*Figure 3c*), but showed significantly reduced growth resistance to the test drugs including doxorubicin, FK506, and valinomycin (*Figure 3d*). As drug transport was severely impaired, we further investigated the mutant's ATPase activity. In presence of transport substrates such as FK506, valinomycin, or verapamil, the ATPase activity of WT Pgp was significantly stimulated; however, the G72A mutant displayed only low levels of basal ATPase activity (in the absence of drug) and the response to drug stimulation was greatly diminished over a range of concentrations tested (*Figure 3e–g*). To reveal potential structural consequences responsible for the loss of function, we performed cryoEM on the Pgp335 G72A mutant after crosslinking the single cysteine L335C with AAC-DNTP and stabilizing the OF complex with $Mg^{2+}$ ATP and vanadate in a post-hydrolysis intermediate state. After multi-model classification, two conformations emerged. The first conformation, resolved at 3.0 Å, shows no substrate bound in analogy to OF335-nolig. The second dominant conformation is, however, much more heterogenous, resulting in a reduced and anisotropic resolution of approximately 4.6 Å (*Figure 3—figure supplement 1*). Here, the NBDs are well resolved but several transmembrane helices are missing, including TM1 in which residue G72A is located. The structural disorder, observed

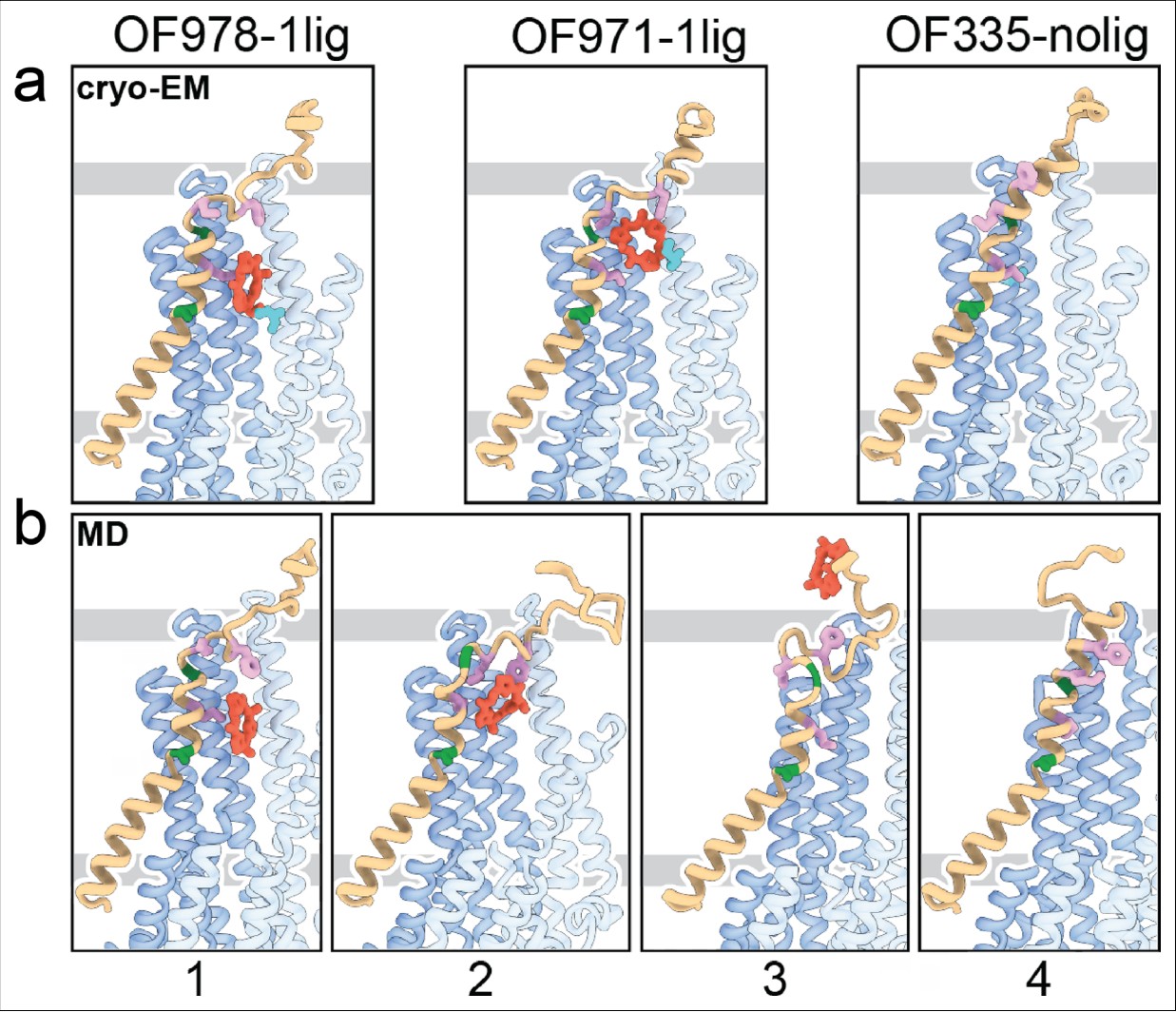

**Figure 4.** Tracing the substrate movement in P-glycoprotein (Pgp). Structures of outward-facing (OF) Pgp from cryoEM (**a**) aligned with four snapshots from molecular dynamics (MD) simulations (**b**) of substrate transport and release. TM1 is rendered in brown, with A63 and G72 in green, and M68, M74, and F78 in plum. The cyclic peptide is shown as red sticks. Significant conformational changes were observed for TM1 along with the upward movement and release of the substrate from OF Pgp.

The online version of this article includes the following figure supplement(s) for figure 4:

**Figure supplement 1.** Snapshots for molecular dynamics (MD) simulation started from OF978-1lig at 400 K.

**Figure supplement 2.** Ligand release in molecular dynamics (MD) simulation starting from OF978-1lig.

**Figure supplement 3.** Molecular dynamics (MD) simulation of ligand release starting from OF335-1lig at T=330 K.

in the transmembrane region, is likely a result of the tethered ligand, causing local distortion of the surrounding α-helices as TM1 may not be able to kink at G72 in the mutant, which could explain its impaired transport function. Collectively, the significant differences in drug resistance profiles and drug-stimulated ATP hydrolysis rates between the G72A mutant and WT-Pgp along with our structural findings substantiate and highlight the key role of TM1 for the regulation of substrate transport in Pgp.

## Discussion

Our data suggest a mechanism for how Pgp mediates substrate transport. Transport commences in the IF conformation with binding of the substrate. Upon ATP-induced NBD dimerization, the transporter converges into an OF conformation in which the intracellular gate is closed. The presence of

the substrate in the OF conformation of Pgp dislocates TM1 at A63 and introduces a sharp kink at G72, forcing the helix to unwind partially between TM1 and TM2 (*Figure 4* and *Figure 2—figure supplement 3*). This enlarges the extracellular loop which shields the transmembrane tunnel from the extracellular environment and guides the substrate to move up the translocation tunnel, allowing TM1 to gradually restabilize. During the last stages of transport (OF971-1lig to OF335-nolig), the TM1-TM2 regulatory loop straightens, and the substrate-interacting hydrophobic residues (M74 and F78) rotate and pivot away from the tunnel, which facilitates the escape of the substrate from the transporter. Although OF971-1lig captured a post-hydrolysis state, we cannot rule out the possibility of ligand release prior to ATP hydrolysis, when it is not covalently bound. Notwithstanding, the proposed TM1 kinking and straightening mechanism of substrate expulsion would obviate the requirement for the Y-shaped, wide-open OF conformation reported for multiple bacterial ABC transporters (*Ward et al., 2007*; *Dawson and Locher, 2006*), suggesting a substrate-dependent transport mechanism in Pgp, or else a divergence of mechanisms among ABC transporters.

For the OF335-2lig structure, observed in the same dataset as OF335-1lig and OF335-nolig, we hypothesize that the hydrophobic nature of the substrate promotes the binding of two substrate molecules to each other. MD simulations with this structure as a starting conformation reveal no separation of the two molecules, suggesting that they might be treated as one bigger substrate in the transport event. With the binding positions differing between OF335-1lig and OF335-2lig (*Figure 2— figure supplements 4c and 7*, *Figure 3—figure supplement 1*), neither the cryoEM structure nor the simulations allow conclusions about a possible transport mechanism of two ligands at this point.

How ABC transporters move substrates through the lipid bilayer has been a long-standing question (*Fan et al., 2020*). The scarcity of high-resolution structures of substrate-bound OF conformations (*Chaptal et al., 2022*), especially in the highly transient intermediate states, has severely limited our understanding of this vital process. Here, we were able to determine eight high-resolution structures of the archetypical ABC transporter Pgp, revealing the underlying molecular mechanics of substrate translocation. In general, the large number and diversity of ABC transporters, and the even wider variety of their substrates, make it unlikely that a universal substrate translocation mechanism will emerge. Our strategy of stalling the otherwise highly transient intermediates in this process through crosslinking and subsequent high-resolution cryoEM analysis could serve as a blueprint to understand the transport mechanisms of lipid transporters as well as other members of the transporter family.

## Materials and methods
### Synthesis of compounds and analysis
The synthesis route and compound numbers are shown in *Figure 1—figure supplement 1*. All organic reactions were carried out under anhydrous conditions and argon atmosphere, unless otherwise noted. Reagents were purchased at the highest commercial quality and used without further purification. NMR spectra were acquired using Bruker DRX-300 or Bruker AV-600 instruments, and chemical shifts ($\delta$) are reported in parts per million (ppm), which were calibrated using residual undeuterated solvent as an internal reference. High-resolution mass spectra (HRMS) were recorded on an Agilent Technologies 6230 TOF LC/MS with a Dual AJS ESI ion source. For flash column chromatography, the stationary phase was 60 Å silica gel.

### General procedure for the synthesis of *N*-Boc-(*S*)-amino thioamides 1 and 2
To a solution of *N*-Boc-(*S*)-amino acid (1.0 g, 5.20 mmol) in THF (20 mL) at 0°C was added triethylamine (0.8 mL, 5.80 mmol) and ethyl chloroformate (1.1 mL, 10.92 mmol). The reaction mixture was stirred for 30 min before the addition of aqueous ammonium hydroxide (20 mL). The reaction mixture was stirred for a further 10 min. The mixture was extracted with ethyl acetate (EtOAc), and the organic layer was washed with brine and dried over anhydrous $Na_2SO_4$, then concentrated under reduced pressure to give *N*-Boc-(*S*)-amino amide as a white solid. The residue was redissolved in THF, Lawesson's reagent (3.0 g, 5.20 mmol) was added, and the reaction mixture was stirred at 50°C for 12 hr. After solvent removal, the residue was redissolved in EtOAc. The organic layer was washed with 1% NaOH, $H_2O$, and brine, then dried over anhydrous $Na_2SO_4$ and concentrated under reduced pressure.

The resulting residue was purified by silica gel column chromatography using EtOAc/hexane (1:1) to give *N*-Boc-(*S*)-amino thioamide **1** or **2** as a yellow solid.

## Synthesis of thiazole ester 3

A solution of thioamide **1** (2.0 g, 9.80 mmol) dissolved in dimethoxyethane (DME, 20 mL) was cooled to –20°C, followed by the addition of $KHCO_3$ (7.84 g, 78.4 mmol) under inert atmosphere. The suspension was stirred for 15 min, followed by the addition of ethyl bromopyruvate (3.6 mL, 29.4 mmol). The reaction mixture was further stirred for 1 hr while it warmed from –20°C to room temperature (rt). The reaction mixture was cooled to –20°C again, and a solution of trifluoroacetic anhydride (5.90 mL, 39.2 mmol) and lutidine (9.7 mL, 83.3 mmol) in DME was added dropwise. The reaction mixture was allowed to warm from 0°C to rt and was stirred at rt for 12 hr. After solvent removal, the residue was redissolved in EtOAc, and the organic layer was washed with brine, dried over anhydrous $Na_2SO_4$, and concentrated under reduced pressure. The resulting residue was purified by silica gel column chromatography using EtOAc/hexane (1:1) as the mobile phase, affording the thiazole ester **3** (2.0 g, 69%) as a yellow solid. MS (ESI) m/z 301 [M+H]$^+$; HRMS: calcd for $C_{13}H_{20}N_2O_4S$ (M+H)$^+$ 301.1143 found 301.1213. $^1$H NMR (600 MHz, CDCl$_3$) δ 8.07 (s, 1H), 5.22 (s, 1H), 5.11 (s, 1H), 4.41 (q, J=7.1 Hz, 2H), 1.62 (d, J=6.9 Hz, 3H), 1.44 (s, 9H), 1.39 (t, J=7.1 Hz, 3H). $^{13}$C NMR (150 MHz, CDCl$_3$) δ 161.50, 147.36, 61.58, 28.45, 21.94, 14.51.

## Synthesis of thiazole ester 4

**4** was synthesized from thioamide **2** (1.0 g, 2.80 mmol) according to the same procedure described for the synthesis of compound **3**. Thiazole ester **4** (1.0 g, 79%) was obtained as a yellow solid. MS (ESI) m/z 453 [M+H]$^+$; HRMS: calcd for $C_{21}H_{28}N_2O_5S_2$ (M+H)$^+$ 453.1439 found 453.1524. $^1$H NMR (600 MHz, CDCl$_3$) δ 8.10 (s, 1H), 7.20 (d, J=8.6 Hz, 2H), 6.83 (d, J=8.6 Hz, 2H), 5.58 (s, 1H), 5.21 (s, 1H), 4.41 (q, J=7.1 Hz, 2H), 3.79 (s, 3H), 3.53 (s, 2H), 3.17–2.95 (m, 2H), 1.46 (s, 9H), 1.39 (d, J=7.1 Hz, 3H). $^{13}$C NMR (150 MHz, CDCl3) δ 161.38, 158.93, 147.52, 130.23, 129.72, 127.73, 114.16, 61.62, 55.40, 52.43, 36.44, 36.20, 28.43, 14.50, 14.26.

## Synthesis of dipeptide 8

To a solution of compound **3** (0.45 g, 1.46 mmol) in dichloromethane (DCM, 10 mL) was added trifluoroacetic acid (TFA, 3.3 mL). The reaction mixture was stirred at rt for 1 hr. The solvent was removed under reduced pressure to give the thiazole amine **5** in quantitative yield. Compound **3** (0.50 g, 1.66 mmol) was dissolved in a mixture of THF/MeOH/$H_2O$ (3:1:1). Then, NaOH (0.532 g, 13.3 mmol) was added, and the reaction mixture was stirred at rt for 1 hr. After removing the solvent at reduced pressure, the residue was redissolved in EtOAc. The organic layer was washed with brine, dried over anhydrous $Na_2SO_4$, and concentrated in vacuo to give thiazole acid **6**. The resulting thiazole acid **6** (0.40 g, 1.46 mmol) was dissolved in anhydrous DMF (5 mL). HOBt (0.670 g, 4.38 mmol), HBTU (1.66 g, 4.38 mmol), and thiazole amine **5** (1.50 mmol, dissolved in DMF) were added sequentially. To this mixture was added iPr$_2$EtN (1.25 mL, 7.2 mmol) with stirring at rt under N$_2$ for 12 hr. Upon completion, the reaction was quenched with aqueous HCl (1 M). The solution was diluted with EtOAc, washed with saturated NaHCO$_3$ solution and brine, dried over Na$_2$SO$_4$, and filtered. The solvent was then removed in vacuo, and the residue was subjected to column chromatography on silica gel to provide the dipeptide **8** (0.560 g, 84%) as a white solid. MS (ESI) m/z 455 [M+H]$^+$; HRMS: calcd for $C_{19}H_{26}N_4O_5S_2$ (M+H)$^+$ 455.1344 found 455.1422. $^1$H NMR (600 MHz, CDCl$_3$) δ 8.09 (s,1H), 8.03 (s, 1H), 7.86 (dd, J=8.5, 3.4 Hz, 1H), 5.59 (tt, J=8.7, 6.1 Hz, 1H), 5.10 (d, J=37.7 Hz, 2H), 4.41 (q, J=7.1 Hz, 2H), 1.79 (d, J=7.0 Hz, 2H), 1.61 (d, J=6.8 Hz, 2H), 1.45 (s, 9H), 1.39 (t, J=7.1 Hz, 3H). $^{13}$C NMR (150 MHz, CDCl$_3$) δ 173.18, 173.15, 160.60, 149.23, 147.30, 147.28, 127.59, 127.58, 123.98, 61.61, 47.27, 47.22, 28.46, 21.19, 21.16, 14.50.

## Synthesis of linear tripeptide 10

Reaction of dipeptide **8** (0.50 g, 1.10 mmol) with TFA (3.3 mL) in DCM (15 mL) afforded the amine **9** in quantitative yield. Compound **4** (0.50 g, 1.10 mmol) was dissolved in a mixture of THF/MeOH/$H_2O$ (3:1:1). NaOH (0.40 g, 8.84 mmol) was added, and the reaction mixture was stirred at rt for 1 hr. After removing the solvent at reduced pressure, the residue was redissolved in EtOAc. The organic layer was washed with brine, dried over anhydrous $Na_2SO_4$, and concentrated in vacuo to give thiazole acid

**7**. The resulting thiazole acid **7** (0.40 g, 0.94 mmol) was dissolved in anhydrous DMF (5 mL), then HOBt (0.431 g, 2.82 mmol), HBTU (1.06 g, 2.82 mmol), and dipeptide amine **9** (1.10 mmol, dissolved in DMF) were added. To this mixture was added iPr$_2$EtN (1.25 mL, 7.2 mmol) with stirring at rt under N$_2$ for 12 hr. Upon completion, the reaction was quenched with aqueous HCl (1 M). The solution was diluted with EtOAc, washed with saturated NaHCO$_3$ solution and brine, dried over Na$_2$SO$_4$, and filtered. The solvent was then removed in vacuo, and the residue was subjected to column chromatography on silica gel to provide the linear tripeptide **10** (0.430 g, 60%) as a white solid. MS (ESI) m/z 761 [M+H]$^+$; HRMS: calcd for C$_{33}$H$_{40}$N$_6$O$_7$S$_4$ (M+H)$^+$ 761.1841 found 761.1911. $^1$H NMR (600 MHz, CDCl$_3$) δ 8.10 (s, 1H), 8.08 (s, 1H), 8.06 (s, 1H), 7.89 (d, J=8.2 Hz, 1H), 7.70 (d, J=8.1 Hz, 1H), 7.16 (d, J=8.6 Hz, 2H), 6.80 (d, J=8.6 Hz, 2H), 5.58 (tt, J=15.2, 7.1 Hz, 2H), 5.44 (s, 1H), 5.17 (s, 1H), 4.41 (q, J=7.1 Hz, 2H), 3.76 (s, 3H), 3.60–3.53 (m, 2H), 3.00 (td, J=12.0, 5.5 Hz, 2H), 1.79 (dd, J=7.0, 3.9 Hz, 6H), 1.46 (s, 9H), 1.39 (t, J=7.1 Hz, 3H). $^{13}$C NMR (150 MHz, CDCl$_3$) δ 173.20, 172.77, 160.54,149.24, 147.28, 130.17, 129.49, 127.59, 124.59, 124.33, 114.17, 61.62, 55.39, 47.31, 47.28, 38.77, 36.27, 28.44, 21.22, 21.17, 14.50.

## Synthesis of cyclic peptide 11

Linear tripeptide **10** (0.20 g, 0.26 mmol) was treated with NaOH (0.084 g, 2.10 mmol) in THF/MeOH/H$_2$O (3:1:1) to hydrolyze the ethyl ester, then with TFA (1 mL) dissolved in DCM to remove the *N-t-*Boc protective group. The residue (0.164 g, 0.25 mmol) was dissolved in a mixture of DMF/DCM (2:1, 30 mL), then a solution of PyBop (0.299 g, 0.57 mmol) and 4-dimethylaminopyridine (0.143 g, 1.17 mmol) in DMF/DCM (2:1, 44 mL) was added slowly over 10 hr using a syringe pump. The reaction mixture was washed with aqueous HCl (1 M), saturated NaHCO$_3$, and brine, and dried over Na$_2$SO$_4$. The solvent was then removed in vacuo. The residue was purified by column chromatography on silica gel to give cyclic peptide **11** (0.102 g, 58%) as a white solid. MS (ESI) m/z 615 [M+H]$^+$; HRMS: calcd for C$_{26}$H$_{26}$N$_6$O$_4$S$_4$ (M+H)$^+$ 615.0898 found 615.0950. $^1$H NMR (600 MHz, CDCl$_3$) δ 8.70 (d, J=7.9 Hz, 1H), 8.65 (dd, J=13.4, 7.7 Hz, 2H), 8.18 (s, 1H), 8.16 (s, 1H), 8.15 (s, 1H), 7.28 (d, J=8.6 Hz, 2H), 6.85 (d, J=8.6 Hz, 2H), 5.68–5.54 (m, 3H), 3.79 (s, 3H), 3.78–3.72 (m, 2H), 3.13 (dd, J=13.8, 4.8 Hz, 1H), 2.84 (dd, J=13.9, 8.4 Hz, 1H), 1.73 (d, J=6.8 Hz, 6H). 13C NMR (150 MHz, CDCl$_3$) δ 171.46, 171.44, 159.74, 158.96, 149.04, 148.92, 130.36, 129.82, 124.78, 124.40, 124.17, 114.17, 114.13, 55.43, 53.57, 51.12, 47.57, 47.39, 38.19, 36.27, 25.13.

## Synthesis of AAC-DNPT

To a solution of cyclic peptide **11** (50 mg, 0.08 mmol) in DCM (2 mL) was added 2,4-dinitrobenzenesulfenyl chloride (22 mg, 0.09 mmol) and TFA (0.018 mL, 0.24 mmol). The reaction mixture was stirred at rt for 1 hr. Reaction was quenched with H$_2$O. The organic layer was washed with brine, dried over anhydrous Na$_2$SO$_4$, and the solvent was then removed in vacuo. The residue was purified by column chromatography on silica gel to give AAC-DNPT (32 g, 57%) as a yellow solid. MS (ESI) m/z 693 [M+H]$^+$; HRMS: calcd for C$_{24}$H$_{20}$N$_8$O$_7$S$_5$ (M+H)$^+$ 693.0058 found 693.0143. $^1$H NMR (600 MHz, CDCl$_3$) δ 9.08 (d, J=2.3 Hz, 1H), 8.71 (d, J=7.5 Hz, 1H), 8.61 (dd, J=15.0, 7.5 Hz, 2H), 8.49 (d, J=9.0 Hz, 1H), 8.42 (dd, J=9.0, 2.4 Hz, 1H), 8.25 (s, 1H), 8.17 (s, 1H), 8.14 (s, 1H), 5.83 (dt, J=7.5, 5.9 Hz, 1H), 5.68–5.58 (m, 2H), 3.45 (d, J=5.8 Hz, 2H), 1.75 (t, J=6.8 Hz, 6H). $^{13}$C NMR (150 MHz, CDCl$_3$) δ 171.85, 159.99, 159.66, 149.61, 148.87, 148.18, 145.83, 145.53, 145.29, 128.91, 127.63, 125.10, 124.91, 124.34, 121.77, 50.87, 47.55, 44.91, 25.08.

## Cell lines

*Pichia pastoris* strain KM71H (aox1_ARG4, arg4) was purchased from Invitrogen (now Thermo Fisher) and was transformed with the Zeocin-resistant expression vector pPICZ containing the mouse *Abcb1a* gene, accession number NM_011076, GenBank JF83415. *S. cerevisiae* JPY201 (MATa ura3 Δste6::HIS3) cells were transformed with the pVT-CL-*Abcb1a* shuttle vector carrying an ampicillin resistance for propagation in *E. coli* and an URA3 selection marker for yeast.

## Expression and purification of single-cysteine mutants of Pgp

Single-Cys mutant constructs of murine Pgp (*Abcb1a*, accession number NM_011076, GenBank JF83415) were generated on a Cysless Pgp (CL-Pgp) background in the *P. pastoris* pPIC-CL-*Abcb1a* expression vector (*Swartz et al., 2014*) by site-directed mutagenesis. For cryoEM structural studies of L335C and V978C, we further substituted the catalytic carboxylates in both NBDs to glutamines,

E552Q/E1197Q, to generate ATP hydrolysis-deficient mutants. The Pgp construct used in this study contained C-terminal hexahistidine and Twin-Strep purification tags (*Zoghbi et al., 2017*). Large-scale Pgp biomass production in *P. pastoris* and microsomal membrane preparations were conducted according to published protocols (*Swartz et al., 2014*; *Bai et al., 2011*).

For ATPase activity measurement and MS analysis, we purified Pgp in the presence of *n*-dodecyl-D-maltopyranoside (DDM) supplemented with the lipid 1-palmitoyl-2-oleoyl-*sn*-glycero-3-phosphoethanolamine (POPE). Briefly, microsomes were resuspended in 20 mM Tris (pH 8.0), 20 mM imidazole, 20% glycerol, 500 mM NaCl, protease inhibitors (10 µg/mL leupeptin and pepstatin A, 2.5 µg/mL chymostatin,1 mM PMSF), 0.2 mM tris(2-carboxyethyl)phosphine with 1% DDM for 60 min at 4°C. After centrifugation at 38,000×*g* for 30 min, Pgp was purified from the supernatant using Ni-NTA affinity chromatography in Buffer A (50 mM Tris pH 8.0, 150 mM NaCl, 20% glycerol) supplemented with 0.067% DDM, 0.04% sodium cholate, and 0.1 mg/mL POPE, and with 20 mM imidazole for wash buffer or 200 mM imidazole for elution buffer. The eluate from Ni-NTA was concentrated for further purification by size exclusion chromatography on a Superdex 200 Increase 10/300 column using 20 mM Tris pH 7.5, 150 mM NaCl, 0.067% DDM, 0.04% sodium cholate, and 0.1 mg/mL POPE.

For cryoEM structural determination, we purified Pgp in a mixture of lauryl maltose neopentyl glycol (LMNG) and cholesteryl hemisuccinate (CHS). Briefly, after the solubilization of microsomes in DDM, Pgp was purified from the supernatant using Ni-NTA affinity chromatography in Buffer A supplemented with 0.02% LMNG and 0.004% CHS. The eluate from Ni-NTA was applied to pre-equilibrated Strep-Tactin Superflow resin and incubated at 4°C for 1 hr. Flow-through was removed, and the resin was washed with Buffer A in the presence of 0.02% LMNG and 0.004% CHS. Pgp was eluted with the same buffer containing 2.5 mM desthiobiotin. The eluate from Strep-Tactin was concentrated for further purification by size exclusion chromatography on a Superdex 200 Increase 10/300 column using a detergent-free buffer containing 20 mM Tris pH 7.5 and 200 mM NaCl.

## Crosslinking between single cysteine mutants of Pgp and AAC-DNPT

The covalent Pgp complexes were typically prepared by the reaction of Pgp (3–5 mg/mL) with 10-fold excess of AAC-DNPT at rt for 30 min. L971C labeling was conducted in the presence of MgATP (10 mM). The DNPT color formation was visualized by eye or monitored by UV-visible spectroscopy ($\lambda_{max}$ = 408 nm, $\varepsilon$=13,800 M$^{-1}$ cm$^{-1}$). For mass spectrometric analysis, AAC-labeled Pgp sample was passed through a PD-10 Sephadex G-25 desalting column to remove excess ligand, and the eluate was further treated by addition of cysteine (5 mM) before performing trypsin digestion.

## ATPase activity assay

ATPase activity of Pgp, with or without AAC labeling, was measured at 37°C using an enzyme-coupled ATP regeneration system (*Urbatsch et al., 1995b*). Briefly, 1 µg Pgp was added to 100 µL of ATP cocktail (50 mM Tris, pH 7.5, 12 mM MgCl$_2$, 6 mM phosphoenolpyruvate, 1 mM NADH, 10 units lactate dehydrogenase, 10 units pyruvate kinase, and 10 mM ATP). The rate of ATP hydrolysis was determined by the decrease in NADH absorbance at 340 nm using a microplate reader (Filtermax F5). Verapamil was added from stocks in water, QZ-Ala, AAC-DNPT, FK506, and valinomycin were added from stocks in DMSO such that the final DMSO concentration was ≤1%. ATPase activity was calculated as described previously (*Urbatsch et al., 2003*). To analyze the activities of Pgp mutants (L335C, V978C, and L971C, with CL-Pgp as control) with varying concentrations of AAC-DNPT (0–20 µM), we incubated Pgp with AAC-DNPT for 15 min at rt prior to the addition of ATP to initiate the ATPase reaction.

## Trypsin digestion of Pgp

Ten µL of 5 mg/mL Pgp, with or without AAC labeling, was added to an S-Trap micro column (Farmingdale, NY, USA). Then, 15 µL of 100 mM triethylammonium bicarbonate (TEAB) buffer (pH 7.5) containing 10% sodium dodecyl sulfate was drawn and mixed with Pgp protein by pipette. Then, 2.5 µL of 10% (vol/vol) H$_3$PO$_4$ in water was added to the S-Trap. After 10 min incubation, 165 µL of binding solution (100 mM TEAB in MeOH/H$_2$O 9:1 [vol/vol], pH 7.1) was added to the acidified Pgp protein. After 10 min incubation, the S-Trap was seated in a 1.5 mL tube and centrifuged at 4000×*g* for 2 min until all solution had passed through the S-Trap membrane. The flow-through liquid was drawn back and centrifuged again. After addition of 150 µL of binding solution, the S-Trap was centrifuged at 4000×*g*

for 2 min to wash the protein. The flow-through liquid was removed, and this washing procedure was repeated three times. Trypsin (4 μg, at an enzyme/protein ratio of 1:12.5, wt/wt) was added to the S-Trap and mixed well. The S-Trap was capped loosely to limit evaporation, then incubated in a dry incubator at 37°C for overnight digestion. After digestion was completed, the resulting peptides were eluted by adding 40 μL of $H_2O$ containing 0.2% formic acid (FA), and centrifuging at 4000×$g$ for 2 min. The flow-through liquid was transferred back to be centrifuged again. Then 35 μL of $CH_3CN/H_2O/FA$ (80:20:0.2%) was added, and the S-Trap was centrifuged at 4000×$g$ for 2 min. The eluted peptides were collected for LC-MS analysis.

## LC-MS analysis

The LC-MS setup consisted of an ultraperformance liquid chromatography instrument (UPLC, Waters, Milford, MA, USA) coupled with a high-resolution Orbitrap Q Exactive mass spectrometer (Thermo Scientific, San Jose, CA, USA). A reversed-phase column (BEH C18, 1.0×100 mm², 1.8 μm) was used for separation. The injection volume was 5 μL per analysis. For gradient elution (0–95% $CH_3CN$ with 0.1% FA in water), the mobile phase flow rate was 30 μL/min. The Orbitrap mass spectrometer was equipped with a heated electrospray ionization source. The rate of sheath gas flow was 10 L/hr and the applied ionization voltage was +4 kV. The ion transfer inlet capillary temperature was kept at 250°C. Mass spectra were acquired using Thermo Xcalibur (3.0.63) software. The scan mode was set to full scan MS (*Juliano and Ling, 1976*), followed by data-dependent MS (*Ueda et al., 1986*) acquisition. The resolution of full scan MS (*Juliano and Ling, 1976*) was 70k and the automatic gain control (AGC) target was set to 5e5. For MS (*Ueda et al., 1986*) acquisition, the resolution was 17.5k, and AGC target was 2e4. The 20 most abundant ions (+2 to +6 ions) were selected to fragment with a normalized collision energy of 30%.

## MDCK-ABCB1 transport assay

The Madin Darby Canine Kidney (MDCK) epithelial cells stably transfected with the human *ABCB1* gene forms a confluent monolayer, which is widely adopted to evaluate if a compound is subject to Pgp efflux based on the permeability measurement in both directions (*Brouwer et al., 2013*). The MDCK-ABCB1 permeability assay for QZ-Ala was conducted by Bioduro-Sundia Inc (San Diego, CA, USA). Briefly, 5 μM QZ-Ala in the absence or presence of 10 μM cyclosporin A was added to either the apical (A) or the basolateral (B) side and the amount of permeation was determined on the other side of the monolayer by LC-MS/MS. The efflux ratio ($R_E = P_{app}$ (B-A) to $P_{app}$(A-B)) for QZ-Ala was determined following standard protocols. An $R_E$ >2.0 generally indicates a substrate for Pgp.

## G72A mutagenesis and drug resistance assays

G72A mutation was conducted on the mouse CL-Pgp template in the pVT expression vector (pVT-CL-*Abcb1a*) (*Swartz et al., 2014*). First, the three N-glycosylation sites N82/N87/N90 that were previously substituted by Gln for X-ray crystallography were restored to the original codons 5'- <u>AAC</u>gtgtccaa g<u>AAC</u>agtact<u>AAT</u>-3' by QuickChange site-directed mutagenesis. The G72A mutation was then added by a second round of site-directed mutagenesis, and the full-length open reading frame sequenced to confirm no other unwanted mutation was present. Plasmids from three individual clones, together with WT-Pgp and CL-Pgp as well as pVT 'empty' vector controls were transformed into *S. cerevisiae* JPY201 (MATa ura3 Δste6::HIS3) cells for expression and functional assays that were performed essentially as previously described (*Swartz et al., 2014*). Briefly, 10 mL yeast cultures were grown overnight in uracil-deficient minimal medium, diluted to OD$_{600}$=0.05 in YPD medium (1% [wt/vol] yeast extract/2% [wt/vol] peptone/2% [wt/vol] glucose), and seeded into 96-well plates containing YPD alone or YPD plus 40 μM doxorubicin, 50 μM FK506, or 100 μM valinomycin. Samples were grown in triplicate wells at 30°C for up to 40 hr, and yeast cell growth was monitored by measuring the OD$_{600}$ at 2 hr increments in a microplate reader (Benchmark Plus, Bio-Rad). The remainder of the 10 mL cultures was used to assess Pgp expression by western blot analysis of microsomal membrane preparations using the monoclonal C219 anti-Pgp antibody (Thermo Fisher, Catalog number MA1-26528).

For ATPase assays, the three N-glycosylation sites were restored in the *P. pastoris* pPIC-CL-*Abcb1a* expression vector and the G72A mutant introduced by site-directed mutagenesis. For cryoEM, L335C was added to the G72A mutant by mutagenesis; the integrity of the open reading frame was confirmed by DNA sequencing after each round of mutagenesis. G72A and G72A/L335C mutant proteins were

purified from *P. pastoris* microsomal membranes as described for single Cys mutants. Bars in *Figure 3* represent the mean of ≥3 independent experiments ± SEM; two-way ANOVA with post hoc Bonferroni tests identified those pairs with very highly significant differences (p<0.001).

## EM sample preparation

After shipment, quality control of the samples was performed by collecting negative stain EM images on a Tecnai G$^2$ Spirit TWIN TEM (FEI) (*Gewering et al., 2018*). For cryoEM, all samples were adjusted to 3.5 mg/mL in 50 mM Tris, pH 7.5, and 200 mM NaCl. OF335 was obtained after reaction with a 4-fold excess of AAC-DNPT; OF978C and 971C were obtained with 10-fold molar excess for 30 min at rt. To trigger NBD dimerization, the QQ constructs were then incubated with 5 mM MgATP for 1 hr at rt prior to grid preparation. For OF971, 5 mM Mg$^{2+}$ATP/Vi was added to the sample and incubated for 1 hr at rt. Freezing protocol was followed as previously described (*Januliene and Moeller, 2021*). All samples were vitrified on freshly glow-discharged CF-1.2/1.3 TEM grids (Protochips, USA) with a Vitrobot Mark IV (Thermo Fisher Scientific, Inc, USA) at 100% humidity and 4°C, with a nominal blot force of –2 and a blotting time of 12 s. Grids were plunged into liquid ethane and stored in liquid nitrogen until further use.

## EM data acquisition and processing

The datasets for OF978 and OF971 were acquired on a Titan Krios G4, operated at 300 kV, and equipped with a Selectris X imaging filter and a Falcon 4 direct electron detector (all Thermo Fisher Scientific, USA). Datasets were obtained using automation strategies of EPU software v2.13 (Thermo Scientific) at a nominal magnification of ×215,000, corresponding to a calibrated pixel size of 0.573 Å. The camera was operated in electron counting mode, and the data were saved in electron-event representation format. All other datasets were obtained on a Titan Krios G3i (Thermo Fisher Scientific, USA), using automation strategies of EPU 2.9 or newer, equipped with a Gatan BioQuantum K3 Imaging Filter (Gatan, USA) in electron counting mode. The nominal magnification was ×105,000 corresponding to a calibrated pixel size of 0.837 Å. The exposure time for all datasets was ~4 s, and the total dose was 70 e⁻/Å (*Ueda et al., 1986*) (Selectris X-Falcon 4) or 75 e⁻/Å (*Ueda et al., 1986*) (BioQuantum K3). Quality of the data was monitored during collection using cryoSPARC live v3.2.0 and v3.3.1 (*Punjani et al., 2017*). Details about number of collected images and picked particles, as well as number of particles in the final map and resolutions are listed in *Supplementary file 1*.

For data processing for IF335, the initial model was obtained using cryoSPARC v3.2.0. Further processing, including 3D classifications and refinements, was obtained in Relion 3.1 (*Scheres, 2012*). For the final refinement, particles and map were transferred back to cryoSPARC v3.2.0 to run a Non-Uniform Refinement (NUR) (*Punjani et al., 2020*). All other datasets were processed in cryoSPARC v3.3.1. Particles were picked broadly with the blob picker, and the best classes were selected for further processing. Sorting of the particles was achieved by multiple heterogenous refinements and NUR. Global and local CTF refinements (*Rubinstein and Brubaker, 2015*) were performed toward the end of the processing pipeline, followed by another round of NUR. An exemplary processing pathway is provided in *Figure 2—figure supplement 5*. Processing results for all structures are shown in *Figure 2—figure supplement 1*.

For all datasets, the images were repicked with the Topaz (*Bepler et al., 2020*) picker, and this increased the resolution for all three maps from the OF335 dataset. For all other datasets, no improvement of the maps could be achieved with this approach. Density modification with *phenix.resolve_cryo_em* (*Terwilliger et al., 2020*) was carried out using two half-maps together with the FSC-based resolution and the molecular masses of the molecules. This procedure resulted in significant improvement of the map qualities. The density-modified map of OF335-1lig was used for *Figure 2—figure supplement 2*.

## Model building

For all datasets, the structures of Pgp in the inward and outward conformations (PDBID: 4Q9I and 6C0V, respectively) were used as templates. All structures were manually edited in COOT (*Emsley and Cowtan, 2004*) and refined using *phenix.real_space_refine*, in combination with rigid-body refinement (*Afonine et al., 2018*) and several rounds of rebuilding in COOT. A quality check of all structures with MolProbity (*Chen et al., 2010*) indicated excellent stereochemistry with 93.1–98.1% of the

non-glycine and non-proline residues found in the most-favored region, and 0.00–0.09% outliers (all-atom clashscore: 9.29–15.76). Refinement and validation statistics are summarized in *Supplementary file 1*. Figures were drawn with ChimeraX (*Goddard et al., 2018*) and PyMOL (The PyMOL Molecular Graphics System, Version 2.0, Schrödinger, LLC).

## MD simulations

MD simulations were performed starting from the structures OF978-1lig, OF335-1lig, and OF335-2lig embedded in a patch of lipid bilayer. The bound cholesterol hemisuccinate molecules in the cryoEM structures were replaced with cholesterol in the same binding poses. Bound magnesium ions and ATP were also retained. The crosslinked ligands were removed and the crosslinking cysteine was replaced by the amino acid originally present at that position. No attempt was made to model the unresolved linker (residues 626–686). Instead, the C-terminus of the first subunit (residue 625) was N-methylated and the N-terminus of the second subunit (residue 687) was acetylated to imitate the presence of an unstructured loop region connecting the two subunits. The structures were then inserted into a model plasma membrane aligned with the xy plane and solvated using CHARMM-GUI (*Wu et al., 2014*; *Lee et al., 2016*) (composition outer leaflet: 30% CHL, 35% POPC, 35% PSM; composition inner leaflet: 30% CHL, 25% PAPC, 25 POPE, 25% POPS). All protonation states were set according to PROPKA3 (*Søndergaard et al., 2011*; *Olsson et al., 2011*). The resulting simulation systems were placed in rectangular boxes with a size of approximately $12 \times 12 \times 17$ nm$^3$ (*Sharom, 2011*) and around 260,000 atoms each. The ligands were modeled using the CGenFF server (*Vanommeslaeghe and MacKerell, 2012*; *Vanommeslaeghe et al., 2012*) and placed into the solvated systems at their original location as in the cryoEM structures with the crosslink to the protein removed.

All MD simulations were performed with the CHARMM36m force field (*Huang et al., 2017*) (version july2021) including CGenFF parameters (*Vanommeslaeghe et al., 2010*) (version 4.6) using GROMACS (*Páll et al., 2020*; *Abraham et al., 2015*) (version 2021.6). The lengths of covalent bonds involving hydrogen atoms were constrained using LINCS (*Hess et al., 1997*) in all simulations. The simulation systems were energy minimized and subsequently equilibrated in six steps while gradually releasing restraints on the positions of the heavy and backbone atoms of the protein, the lipid atoms, and the ligand atoms (see *Supplementary file 3*).

The production simulations were started from the last configuration of the equilibration with random initial velocities. All production simulations were performed in the NPT ensemble using a semi-isotropic Parrinello-Rahman barostat (*Parrinello and Rahman, 1981*) with a target pressure of 1 bar and a coupling constant of 5 ps. The velocity rescale thermostat (*Bussi et al., 2007*) with a time constant of 1 ps was used to keep the target temperatures of T=330 K and T=400 K, respectively. We used a time step of 0.002 ps in all simulations.

## Acknowledgements

We thank J-H Leung, W-H Lee, and A Ward for early contributions, C Katz for technical assistance, S Welsch and S Prinz for support during the cryoEM data collection, W Kühlbrandt for generous access to the cryoEM facilities in Frankfurt and general support, and W Kühlbrandt and I Wilson for critical reading and editing of the manuscript. This work was supported by National Institute of General Medical Sciences R01 GM118594 and GM148675 (QZ) and GM141216 (ILU), the Deutsche Forschungsgemeinschaft (DFG) Sonderforschungsbereich 944 and 1557 and INST 190/196 (AM), the Bundesministerium für Bildung und Forschung (BMBF) 01ED2010 (AM), the Max Planck Society (HJ, GH, AM, TG), the South Plains Foundation (ILU), and the National Science Foundation CHE-2203284 (HC).

## Additional information

### Funding

| Funder | Grant reference number | Author |
|---|---|---|
| National Institutes of Health | GM118594 | Qinghai Zhang |
| National Institutes of Health | GM148675 | Qinghai Zhang |
| National Institutes of Health | GM141216 | Ina Urbatsch |
| South Plains Foundation | | Ina Urbatsch |
| Deutsche Forschungsgemeinschaft | Sonderforschungsbereich 944 | Arne Moeller |
| Bundesministerium für Bildung und Forschung | 01ED2010 | Arne Moeller |
| National Science Foundation | CHE-2203284 | Hao Chen |
| Deutsche Forschungsgemeinschaft | Sonderforschungsbereich 1557 | Arne Moeller |
| Deutsche Forschungsgemeinschaft | INST 190/196 | Arne Moeller |

The funders had no role in study design, data collection and interpretation, or the decision to submit the work for publication. Open access funding provided by Max Planck Society.

### Author contributions

Theresa Gewering, Investigation, Writing – original draft, TG performed cryoEM and single particle analysis; Deepali Waghray, Investigation, Writing – original draft, DW conducted ligand synthesis and crosslinking, identified Pgp single Cys mutants, performed biochemical characterization, and optimized and prepared Pgp samples for mass spectrometry and cryoEM studies; Kristian Parey, Investigation, KP built molecular models; Hendrik Jung, Investigation, Writing – review and editing, HJ conducted molecular dynamics simulations; Nghi NB Tran, Investigation, NT assayed G72A function; Joel Zapata, Investigation, JZ expressed Pgp mutants; Pengyi Zhao, Investigation, Writing – review and editing, PZ performed mass spectrometric characterization of crosslinking; Hao Chen, Gerhard Hummer, Supervision, Investigation, Writing – review and editing; Dovile Januliene, Investigation, Writing – original draft, DJ assisted with cryoEM experiments and data analysis; Ina Urbatsch, Supervision, Investigation, Writing – original draft, Writing – review and editing, ILU assisted with the mutant design and Pgp labeling experiments; Arne Moeller, Supervision, Funding acquisition, Investigation, Writing – original draft, Writing – review and editing, AM designed and supervised the cryoEM research; Qinghai Zhang, Conceptualization, Supervision, Funding acquisition, Investigation, Writing – original draft, Writing – review and editing, QZ conceived and initiated the project, designed the research, and coordinated collaborations

### Author ORCIDs

Hendrik Jung ![ORCID] https://orcid.org/0000-0002-2159-0391
Hao Chen ![ORCID] http://orcid.org/0000-0001-8090-8593
Dovile Januliene ![ORCID] https://orcid.org/0000-0002-3279-7590
Gerhard Hummer ![ORCID] http://orcid.org/0000-0001-7768-746X
Ina Urbatsch ![ORCID] http://orcid.org/0000-0002-3080-5649
Arne Moeller ![ORCID] http://orcid.org/0000-0003-1101-5366
Qinghai Zhang ![ORCID] https://orcid.org/0000-0003-0866-527X

Reviewer #1 (Public Review): https://doi.org/10.7554/eLife.90174.3.sa1
Reviewer #3 (Public Review): https://doi.org/10.7554/eLife.90174.3.sa2

Author Response https://doi.org/10.7554/eLife.90174.3.sa3

## Additional files

### Supplementary files

- Supplementary file 1. CryoEM data collection, refinement, and validation statistics.
- Supplementary file 2. Summary of cryoEM data for the P-glycoprotein (Pgp) mutants.
- Supplementary file 3. Equilibration protocol for molecular dynamics (MD) simulations.
- MDAR checklist

### Data availability

All cryoEM density maps have been deposited in the Electron Microscopy Data Bank under accession numbers EMD-14754, EMD-14755, EMD-14756, EMD-14758, EMD-14759, EMD-14760, EMD-14761, EMD-15687 and EMD-17630. Atomic coordinates for the atomic models have been deposited in the Protein Data Bank under accession numbers 7ZK4, 7ZK5, 7ZK6, 7ZK8, 7ZK9, 7ZKA, 7ZKB, 8AVY and 8PEE.

The following datasets were generated:

| Author(s) | Year | Dataset title | Dataset URL | Database and Identifier |
|---|---|---|---|---|
| Parey K, Januliene D, Gewering T, Moeller A | 2023 | The ABCB1 L335C mutant (mABCB1) in the outward facing state | https://www.ebi.ac.uk/emdb/EMD-14754 | Electron Microscopy Data Bank, EMD-14754 |
| Parey K, Januliene D, Gewering T, Moeller A | 2023 | ABCB1 L335C mutant (mABCB1) in the outward facing state bound to AAC | https://www.ebi.ac.uk/emdb/EMD-14755 | Electron Microscopy Data Bank, EMD-14755 |
| Parey K, Januliene D, Gewering T, Moeller A | 2023 | ABCB1 L335C mutant (mABCB1) in the outward facing state bound to 2 molecules of AAC | https://www.ebi.ac.uk/emdb/EMD-14756 | Electron Microscopy Data Bank, EMD-14756 |
| Parey K, Januliene D, Gewering T, Moeller A | 2023 | ABCB1 L971C mutant (mABCB1) in the outward facing state bound to AAC | https://www.ebi.ac.uk/emdb/EMD-14758 | Electron Microscopy Data Bank, EMD-14758 |
| Parey K, Januliene D, Gewering T, Moeller A | 2023 | ABCB1 L971C mutant (mABCB1) in the inward facing state | https://www.ebi.ac.uk/emdb/EMD-14759 | Electron Microscopy Data Bank, EMD-14759 |
| Parey K, Januliene D, Gewering T, Moeller A | 2023 | ABCB1 V978C mutant (mABCB1) in the outward facing state bound to AAC | https://www.ebi.ac.uk/emdb/EMD-14760 | Electron Microscopy Data Bank, EMD-14760 |
| Parey K, Januliene D, Gewering T, Moeller A | 2023 | ABCB1 V978C mutant (mABCB1) in the inward facing state | https://www.ebi.ac.uk/emdb/EMD-14761 | Electron Microscopy Data Bank, EMD-14761 |
| Parey K, Januliene D, Gewering T, Moeller A | 2023 | ABCB1 L335C mutant (mABCB1) in the inward facing state bound to AAC | https://www.ebi.ac.uk/emdb/EMD-17630 | Electron Microscopy Data Bank, EMD-17630 |
| Parey K, Januliene D, Gewering T, Moeller A | 2023 | The ABCB1 L335C mutant (mABCB1) in the Apo state | https://www.ebi.ac.uk/emdb/EMD-15687 | Electron Microscopy Data Bank, EMD-15687 |
| Parey K, Januliene D, Gewering T, Moeller A | 2023 | The ABCB1 L335C mutant (mABCB1) in the outward facing state | https://www.rcsb.org/structure/7ZK4 | RCSB Protein Data Bank, 7ZK4 |
| Parey K, Januliene D, Gewering T, Moeller A | 2023 | ABCB1 L335C mutant (mABCB1) in the outward facing state bound to AAC | https://www.rcsb.org/structure/7ZK5 | RCSB Protein Data Bank, 7ZK5 |

*Continued on next page*

*Continued*

| Author(s) | Year | Dataset title | Dataset URL | Database and Identifier |
|---|---|---|---|---|
| Parey K, Januliene D, Gewering T, Moeller A | 2023 | ABCB1 L335C mutant (mABCB1) in the outward facing state bound to 2 molecules of AAC | https://www.rcsb.org/structure/7ZK6 | RCSB Protein Data Bank, 7ZK6 |
| Parey K, Januliene D, Gewering T, Moeller A | 2023 | ABCB1 L971C mutant (mABCB1) in the outward facing state bound to AAC | https://www.rcsb.org/structure/7ZK8 | RCSB Protein Data Bank, 7ZK8 |
| Parey K, Januliene D, Gewering T, Moeller A | 2023 | ABCB1 L971C mutant (mABCB1) in the inward facing state | https://www.rcsb.org/structure/7ZK9 | RCSB Protein Data Bank, 7ZK9 |
| Parey K, Januliene D, Gewering T, Moeller A | 2023 | ABCB1 V978C mutant (mABCB1) in the outward facing state bound to AAC | https://www.rcsb.org/structure/7ZKA | RCSB Protein Data Bank, 7ZKA |
| Parey K, Januliene D, Gewering T, Moeller A | 2023 | ABCB1 V978C mutant (mABCB1) in the inward facing state | https://www.rcsb.org/structure/7ZKB | RCSB Protein Data Bank, 7ZKB |
| Parey K, Januliene D, Gewering T, Moeller A | 2023 | The ABCB1 L335C mutant (mABCB1) in the Apo state | https://www.rcsb.org/structure/8AVY | RCSB Protein Data Bank, 8AVY |
| Parey K, Januliene D, Gewering T, Moeller A | 2023 | ABCB1 L335C mutant (mABCB1) in the inward facing state bound to AAC | https://www.rcsb.org/structure/8PEE | RCSB Protein Data Bank, 8PEE |

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
