## [Editor Report · eLife assessment]

P-glycoprotein is a major ABC-transporter that exports drugs used in chemotherpay and effects the pharmacokinetics of other drugs. Here the authors have determined cryo-EM structures of drug complexes in previously unforeseen outward-facing conformations. These **convincing** findings are mechanistically **important** and reveal potential regions to be exploited by rational-based drug design.

---

## [Referee Report · Reviewer #1 (Public Review)]

Summary

Here the authors have tethered a Pgp substrate to strategically place cysteine residues in the protein. Notably, the cysteine-linked substate (ANC-DNPT)- stimulate ATP hydrolyse and so are able to undergo IF to OF transitions. The authors then determined cryo-EM structures of these complexes and MD simulations of bound states. By capturing unforeseen OF conformations with substate they propose that TM1 undergoes local conformational changes that are sufficient to translocate substrates, rather than large bundle movements.

Strengths: This paper provides the first substrate (ANC-DNPT)- bound conformations of PgP and a new mechanistic model of how substrates are translocated.

Weaknesses: Although the cross-links stimulate ATP hydrolysis, it is unclear if the TM1 conformations are exactly the same under physiological conditions, since they have been covalently-trapped to the substrate.

---

## [Referee Report · Reviewer #3 (Public Review)]

Summary: The authors used cross-linking of a known P-gp substrate in combination with single particle cyro-EM to investigate the translocation pathway of this important ABC transporter. Based on the results of this study, a new translocation mechanism is proposed that is supported by the data. While only one substrate was used, the data obtained are convincing. In addition, the proposed model will stimulate new experiments from other laboratories to proof or disproof the model.

Strengths: the combination of cross-linking and structural biology allowed novel insights in the translocation pathway of ABCB1

Weaknesses: While only one substrate was used, the data obtained are convincing. In addition, the proposed model will stimulate new experiments from other laboratories to proof or disproof the model.

---

## [Author Response]

The following is the authors’ response to the original reviews.

**Reviewer 1 (Public Review):**
Weakness: Although the cross-links stimulate ATP hydrolysis, further controls are needed to convince me that the TM1 conformations observed in the structures are physiologically relevant, since they have been trapped by "large" substrates covalently-tethered by crosslinks.

Our response: Reviewer 1 raised concerns about the relatively large size of our covalently attached AAC substrate that would potentially distort TM1 in Pgp. We would like to clarify that AAC has a molecular weight of 462 Da, which, in comparison to many known Pgp substrates ranging from 250 to over 1,000 Da, is not a large compound. For instance, the few other Pgp substrates mentioned in our manuscript all have a comparable or larger size: verapamil, 455 Da; doxorubicin, 544 Da; FK506, 804 Da; valinomycin, 1,111 Da; cyclosporin A, 1,203 Da.

Furthermore, AAC was strategically attached to a site distant from TM1 in the inwardfacing Pgp conformation. After it was exported to the outward-facing state, several TM helices accommodate the compound. The observation that only TM1 exhibited significant conformational changes suggests its potential role in the transport mechanism. This hypothesis is supported by our findings, where a conservative substitution (G72A) in TM1 resulted in a dramatic loss of transport function for various drug substrates and impaired verapamil-stimulated ATPase activity.

**Reviewer 1 (Recommendations for the Authors):**
I understand the need for an unconventional approach to understanding the translocation pathway. What would help to support this model is to cross-link a much smaller substrate, as the one used is quite large and could potentially distort TM1 in the outward-state when cross-linked.

Our response: We thank the reviewer for this recommendation, and we have outlined plans for future experiments involving other substrates, including smaller ones, to further investigate our proposed model. However, it is important to acknowledge that conducting these studies will require a significant amount of effort and resources, which we believe extend beyond the scope of our current manuscript.

In unbiased MD simulations starting from the IF state are there any simulations where the substrate follows the same path as proposed here?

Our response: All our MD simulations were performed in the outward-facing state to focus on potential substrate release pathways. Starting MD simulations from the inwardfacing state would introduce complexities in capturing the necessary domain motions and nucleotide binding and hydrolysis required for substrate translocations. Therefore, we opted not to perform MD studies starting from the inward-facing state.

**Reviewer 2 (Public Review):**
Weakness: There is much to like about the experimental work here but I am less sanguine on the interpretation. The main idea is to covalently link via disulfide bonds a model tripeptide substrate under different conditions that mimic transport and then image the resulting conformations. The choice of the Pgp cysteine mutants here is critical but also poses questions regarding the interpretation. What seems to be missing, or not reported, is a series of control experiments for further cysteine mutations.

Our response: Reviewer 2 raised concerns about the interpretation of our results and suggested the need for additional mutant designs to validate our proposed TM1 mechanism. Firstly, we believe that the observed TM1 conformational changes are valid in our cryoEM structures, despite the use of different conditions and several mutants to capture Pgp in the outward-facing state.

Regarding the G72A mutant, we consider it conclusive that this single point mutation in the TM1 has a profound effect. Importantly, the G72A mutant was readily expressed and purifiable as a stable protein. We were able to resolve a high-resolution structure of the G72A mutant (without the substrate), confirming that the protein is not generally destabilized but properly folded.

Above all, we appreciate the Reviewer’s suggestion to explore additional mutations and intend to do so in future studies.

**Reviewer 2 (Recommendations for the Authors):**
I am sold on the results regarding TM1 conformational changes as they are evident in the cryoEM structures. However, the set of states compared between mutants are not biochemically equivalent: for 335 and 978 they used an ATP-impaired Pgp whereas for 971 they used what appears to be WT, and the conformation was imaged presumably subsequent to ATP hydrolysis and Vanadate trapping. This is significant if the authors were unable to trap the OF in the impaired mutant background and should be highlighted. I have to believe that they tried that condition but I could be wrong.

Our response: We acknowledge the point made by the Reviewer about the biochemical equivalence of mutant states and the potential significance of using an ATP-impaired mutant for trapping the outward-facing conformation of 971. We have not yet attempted to use the ATPase-deficient 971C mutant for crosslinking and intend to address this question in future studies.

In our current approach, we used the ATPase-active 971C for two specific reasons:

1. Our biochemistry data, as shown in Fig 1C, indicates that 971C only crosslinks in the presence of ATP hydrolysis conditions. Vanadate trapping was employed to stabilize the outward-facing conformation.

2. Based on our experience, we have observed that the conformations of ATP-bound (mutant) and vanadate-trapped states of an ABC transporter are structurally equivalent at this resolution level of our study (see ref. 21: Hoffmann et al. NATURE 2019).

The authors propose a new model for substrate translocation. It is based on three mutants and a number of structures. If the authors were not challenging the current dogma I would not have written the next comment. Considering the impact of the findings, I would have designed a couple more cysteine mutants based on their model. For instance, this pathway has a number of stabilizing interactions, can't they make a mutant that preserves conformational switching but eliminates substrate translocation? I like the G97A mutant result but I am worried that the effect could just be a general destabilization or misfolding as part of the cryoEM particles seem to suggest. The authors advance one interpretation of the disorder observed in this mutant but it could easily be my interpretation.

Our response: We thank the reviewer for the suggestion to design additional mutants to further validate our proposed model for substrate translocation. We agree that this would be highly valuable, considering the potential impact of our findings. However, given the time-intensive nature of our approach, we believe that presenting these additional designs in a future study is a reasonable course of action.

Regarding the G72A mutation, we believe that our current data fully supports our model and the role of TM1 in regulating the Pgp activity. Importantly, we would like to emphasize that the G72A mutant was readily expressed and purifiable as a stable protein. Additionally, our cryoEM structural determination of the G72A mutant at high resolution confirmed that the protein is not generally destabilized but properly folded.

There are a couple of troubling methodological questions that I want the authors to address or clarify:1. In the methods they report that the final sample for cryoEM was prepared on a SEC devoid of detergent. It is obvious that the sample was folded but I was wondering why the detergent was removed? Was that critical for observing these structures with multiple ligands? Did they observe any lipids in their cryoEM?

Our response: We avoid detergent in the buffer on final SEC purification. This step is to remove free detergent from the background which helps during cryoEM imaging. Of course, this cannot be done with every detergent but due to the very low CMC of LMNG it is possible. By now, we have verified this method for several other transporters with the same success. While this procedure helps us to obtain better images it is not necessary to obtain specific conformations or ligand bound states, nor does it affect these states or conformations.

In our cryoEM structures , we did observe multiple cholesterol hemisuccinate (CHS) molecules on the outer transmembrane surface of Pgp.

1. Can the authors comment on why labeling was carried out in the presence of ATP? Does it matter if the substrate was added prior to ATP and incubated for a few minutes?

Our response: For every dataset, we first added the substrate to be cross-linked and afterwards added the ATP. In the cases of 335C and 978C, labeling was successful before ATP was added, as evidenced by the inward-facing structures with cross-linked substrate. However, for 971C, cross-linking only occurred after the addition of ATP. We interpret this data to suggest that the 971 site is inaccessible to the substrate in the inward-facing state, and cross-linking can only occur after the transporter transitions to outward-facing state. This is in line with our inward-facing structure which does not show a cross-linked substrate, and our biochemical data shown in Fig 1C, where 971C only crosslinked in the presence of ATP.

1. I am not an expert on MD simulations and I understand that carrying out simulations at higher temperatures used to be a trick to accelerate the process. Is this still necessary? Why didn't the author use approaches such as WESTPA?

Our response: Most so-called enhanced sampling methods, including WESTPA, explicitly define a reaction coordinate for the process of interest, usually based on intuition or prior studies. If this coordinate is chosen poorly, enhanced sampling usually fails, either because the sampling becomes inefficient or because the sampling biases the transition pathway (or both). Lacking reliable intuition or prior knowledge on which motions would result in substrate release, we chose temperature to speed up the process. High temperature largely avoids the introduction of an any bias through the definition of a progress coordinate. By contrast, the weighted ensemble method underlying WESTPA is a great method to simulate unbiased dynamics of a process with a known progress coordinate, but unfortunately requires to choose a progress coordinate prior to the simulation and will then mostly sample the process along this progress coordinate, because this is the only direction in which sampling is improved. High temperature MD on the other hand accelerates all processes in the system under study. Indeed, we have now confirmed that the pathway found at high temperature is also feasible at near-ambient conditions.

In new simulations, we have now observed a similar release pathway at T=330 K. As the only difference, the substrate has not fully dissociated from the protein after 2.5 us, with weak interactions persisting at the top part of TM1 from the extracellular side. Importantly, this is a configuration observed also in higher temperature simulations but with much shorter lifetime.

In response, we now included these new findings and a new Extended Data Fig. 15 in the revised manuscript.

1. One way to show that the two substrates binding mode is biochemically relevant is to measure Vmax at different substrate concentrations. One would expect a cooperative transition if that interaction is mechanistically important.
**Reviewer 3 (Public Review):**

We thank Reviewer 3 for recommending the acceptance of our manuscript as is.

**Reviewer 3 (Recommendations for the Authors):**
Page 4, last line: Pgp302 should be Pgp1302. In addition, I can only encourage the authors to add an additional table to the manuscript. Here, the mutation, the obtained structure(s), IF or OF, the resolution, and the main message should be summarized.

Our response: Following the reviewer’s suggestion, we have added Extended Data Table 2 summarizing the Pgp mutants and respective structural data in the revised manuscript.

We verified that Pgp302 is the correct term on Page 4, last line.

Pg. 5, section 'Covalent ligand design for Pgp labeling', it is mentioned that even in the presence of Mg2+ATP, Pgp302 could not react with AAC-DNPT. Maybe it would be worthwhile to add the data either in Supplementary Information or state 'data not shown'.

Our response: We stated ‘data not shown’ in the text.

Pg. 47, last line : A space is missing between M68, and M74.

Our response: Space was added.

Pg. 7, line 2: The authors mention that a single dataset of ATP-bound Pgp335 revealed three different OF conformations: ligand-free, single-ligand-bound, and double-ligandbound. However, the percentage fraction of each dataset sums up to be more than 100%. Would request the authors to recalculate the fraction size of each conformation.

Our response: We have corrected the error in our calculation, based on the particle distribution in our dataset (OF335-nolig: 1,437,110 particles, 40.4%; OF335-1lig: 1,184,253 particles, 33.3%; and OF335-2lig: 939,924 particles, 26.4%).

Pg 53, Figure legend of Extended Data Fig. 11: Please include the color coding for the helix TM1 and also the residues colored plum.

Our response: We added the color coding for TM1 and other residues in the figure legend.

Pg. 8, line 3: While referring to the structure of OF971-1lig, the authors nicely point towards the conserved residues M74 and F78 which coordinate the ligand. However, in Fig. 3b, residues M74 and F78 should also be indicated.

Our response: We updated Fig. 3b by adding arrows pointing towards the residues M74 and F78.

Pg. 54, Extended data Fig. 12: The authors should adopt a single writing style. In some places, Pgp is referred to as P-gp while in others as Pgp.

Our response: We updated the protein labels in Extended Data Fig. 12.

Pg. 54, Extended data Fig. 12: The authors should clearly mention which OF335 structure (1st panel) was used for visualizing the interactions.

Our response: To clarify, we added the following sentences in the figure legend: “Pgp335 OF in the top panel refers to OF335-1lig. In the bottom panel describing OF335-2lig, the left and right diagrams refer to the binding positions of non-covalent and covalent ligand, respectively”.

Pg. 18, section 'synthesis of dipeptide 8': In the text it is mentioned that for the synthesis of thiazole acid 6, compound 3 was dissolved in a mixture of THF/MeOH/H2O (3:1:1), while in the corresponding figure (Extended Data Fig. 1), the ratio is stated as 5:1:2.

Our response: 3:1:1 ratio is correct. We made the correction in Extended Data Fig. 1.

Pg. 19, section 'synthesis of linear tripeptide 10': Same as above for compounds 10 and 4, respectively.

Our response: We corrected the conditions in the Extended Data Fig. 1 accordingly.

Pg. 20, section 'Synthesis of cyclic peptide 11': There seems to be a discrepancy in the synthesis protocol between the text and the extended figure 1, especially regarding the use of THF/MeOH/H20, followed by NaOH and TFA or only NaOH and TFA.

Our response: we further clarified the conditions of using NaOH in THF/MeOH/H2O (3:1:1) and TFA in DCM in the text for synthesis and Extend Data Fig. 1.

Pg. 40, Extended Data Fig. 1: In the bottom last panel showing the synthesis of peptide 11, the authors have missed showing peptide 10 as the starting material for the reaction.

Our response: Label for the peptide 10 was added following the suggestion.

Pg. 26, third last line: 'o' is missing from the last word cry'o'

Our response: We corrected the typo.

Pg. 63 and 64, Extended Data Table 1: The Cryo-EM data collection, refinement, and validation statistics for OF971-1lig, IF971-1lig, OF978-1lig, and IF978-2lig are mentioned twice in the table.

Our response: This was now corrected in the revision.